# Whole-Genome Sequencing-Based Screening of MRSA in Patients and Healthcare Workers in Public Hospitals in Benin

**DOI:** 10.3390/microorganisms11081954

**Published:** 2023-07-31

**Authors:** Carine Laurence Yehouenou, Bert Bogaerts, Kevin Vanneste, Sigrid C. J. De Keersmaecker, Nancy H. C. Roosens, Arsène A. Kpangon, Dissou Affolabi, Anne Simon, Francis Moise Dossou, Olivia Dalleur

**Affiliations:** 1Clinical Pharmacy Research Group (CLIP), Louvain Drug Research Institute (LDRI), Université Catholique de Louvain UCLouvain, Avenue Mounier 73, 1200 Brussels, Belgium; olivia.dalleur@uclouvain.be; 2Laboratoire de Référence des Mycobactéries (LRM), Cotonou BP 817, Benin; affolabi_dissou@yahoo.fr; 3Faculté des Sciences de la Santé (FSS), Université d’Abomey Calavi (UAC), Cotonou 01 BP 188, Benin; 4Transversal Activities in Applied Genomics, Sciensano, Juliette Wytsmanstraat 14, 1050 Brussels, Belgium; bert.bogaerts@sciensano.be (B.B.); kevin.vanneste@sciensano.be (K.V.); sigrid.dekeersmaecker@sciensano.be (S.C.J.D.K.); nancy.roosens@sciensano.be (N.H.C.R.); 5Ecole Nationale des Techniciens Supérieurs en Santé Publique et Surveillance Épidémiologique, Université de Parakou, Parakou, Benin; docarsene@gmail.com; 6Centre National Hospitalier et Universitaire Hubert Koutoukou Maga (CNHU-HKM), Cotonou BP 386, Benin; 7Centres Hospitaliers Jolimont, Prévention et Contrôle des Infections, Groupe Jolimont Asbl, Rue Ferrer 159, 7100 Haine-Saint-Paul, Belgium; anne.simon@jolimont.be; 8Department of Surgery and Surgical Specialties, Faculty of Health Sciences, Campus Universitaire, Champs de Foire, Cotonou 01 BP 118, Benin; dosfm@yahoo.fr; 9Pharmacy, Clinique Universitaire Saint-Luc, Université Catholique de Louvain (UCLouvain), Avenue Hippocrate 10, 1200 Brussels, Belgium

**Keywords:** methicillin-resistant *Staphylococcus aureus*, healthcare workers, whole-genome sequencing, low- or middle-income countries

## Abstract

Methicillin-resistant *Staphylococcus aureus* (MRSA) constitutes a serious public health concern, with a considerable impact on patients’ health, and substantial healthcare costs. In this study, patients and healthcare workers (HCWs) from six public hospitals in Benin were screened for MRSA. Strains were identified as MRSA using conventional microbiological methods in Benin, and confirmed using matrix-assisted laser desorption/ionization time-of-flight mass spectrometry in Belgium. Whole-genome sequencing (WGS) was used on the confirmed MRSA isolates, to characterize their genomic content and study their relatedness. Amongst the 305 isolates (304 wound swabs and 61 nasal swabs) that were collected from patients and HCWs, we detected 32 and 15 cases of MRSA, respectively. From this collection, 27 high-quality WGS datasets were obtained, which carried numerous genes and mutations associated with antimicrobial resistance. The *mecA* gene was detected in all the sequenced isolates. These isolates were assigned to five sequence types (STs), with ST8 (55.56%, n = 15/27), ST152 (18.52%, n = 5/27), and ST121 (18.52%, n = 5/27) being the most common. These 27 isolates carried multiple virulence genes, including the genes encoding the Panton–Valentine leukocidin toxin (48.15%, n = 13/27), and the *tst* gene (29.63%, n = 8/27), associated with toxic shock syndrome. This study highlights the need to implement a multimodal strategy for reducing the risk of the cross-transmission of MRSA in hospitals.

## 1. Introduction

*Staphylococcus aureus* is an important pathogen in healthcare and community settings that can cause a broad spectrum of diseases, ranging from skin and soft-tissue infections to invasive diseases such as pneumonia, septicemia, and endocarditis [1]. In the 1960s [2], soon after the introduction of methicillin into the healthcare system, methicillin-resistant *S. aureus* (MRSA) was first reported in the United Kingdom. It has since been reported in many countries [3,4] and is known to be a common cause of nosocomial infections (i.e., acquired in a hospital or other healthcare setting) and an important nosocomial pathogen.

Worldwide, the prevalence of healthcare-associated infections (HAIs) in hospitalized patients is approximately 4.5–15.5% [5]. While surgical site infections (SSIs) are the third most common cause of HAIs in high-income countries [6,7], they are the primary source in income-poor countries. *S. aureus* is one of the most frequently isolated bacteria in SSIs [8,9]. A subset of these *S. aureus* strains carries the *mecA* gene, which confers resistance to most beta-lactam antibiotics, including methicillin [10]. The *mecA* gene is usually located on a mobile genetic element called staphylococcal cassette chromosome *mec* (SCCmec). SCCmec typing can be used to distinguish MRSA [11]. MRSA infections present considerable therapeutic challenges that can lead to severe complications (including death) and higher costs incurred by extended hospital stays. Due to their resistance to beta-lactam antibiotics, hospital-acquired MRSA strains present limited treatment options, especially in developing countries [12,13,14]. 

Several techniques are used to study the relatedness of MRSA strains, including pulsed-field gel electrophoresis (PGFE), *spa* typing, multilocus sequence typing (MLST), DNA microarrays, and whole genome sequencing (WGS) [15,16]. The use of these techniques in studying the genomic diversity of MRSA strains has improved our understanding of their origin and transmission into hospitals and communities [17] and has confirmed the infiltration of community-acquired MRSA (CA-MRSA) in healthcare settings [18]. MRSA strains can produce numerous virulence factors, which can be divided into three categories: exo-enzymes, host immunity factors, and toxins. Although, historically, most of these virulence factors have been associated with either CA-MRSA or hospital-associated MRSA (HA-MRSA), numerous exceptions have been reported. For instance, although Panton-Valentine leukocidin (PVL), a phage-borne toxin that causes cell death and tissue necrosis, has historically been linked to CA-MRSA, recent reports have documented PVL-positive HA-MRSA strains in hospitals in Egypt and communities in Tunisia and Algeria [19,20]. These strains also carried genes encoding other toxins, such as toxic shock syndrome toxin-1 (TSST-1), a superantigen that can mediate fever, hypotension, rash, multi-organ dysfunction, and lethal shock; and exfoliative toxin 1 (ETA), a toxin that can lead to hydrolysis in the superficial skin layers, leading to cutaneous infections. 

The nasal carriage of *S. aureus* has been recognized as a risk factor for healthcare-associated infections in different hospital settings, including general populations, surgical patients, and those admitted to intensive care units. Nasal colonization could be utilized as a marker for screening and prevention purposes [21], but infection control and surveillance in low- and middle-income countries (LMICs) are frequently hindered by the limited research on the circulating genotypes of *S. aureus*. This motivates the present study, which, to the best of our knowledge, is the first to focus on MRSA screening in hospitals in Benin. The study is part of a larger project aiming to improve hand hygiene and promote the rational use of antibiotics in the surgical services of these hospitals. Previous results from this project indicated that healthcare workers (HCWs) had a relatively low compliance with hand hygiene protocols in these hospitals [22]. Because patients and HCWs are at the interface between communities and hospitals, we investigated whether cross-transmission occurred and whether HCWs or patients could serve as reservoirs for pathogenic MRSA. 

## 2. Material and Methods

### 2.1. Study Design

This study is a part of the larger MUSTPIC project, covering the gastrointestinal and obstetrics departments of six public hospitals in Benin. These hospitals are located in the southern part of the country, and all perform gastrointestinal and obstetric surgeries. The wound sampling from SSIs in patients was performed from January 2019 to January 2020, and the nasal screening of HCWs was conducted later in June 2020. There is no history of transmission among patients between hospitals. 

### 2.2. Sample Collection

Wound swabs were taken from patients who presented clinical signs of SSI according to the Centers for Diseases Control and Prevention (CDC) definitions [23]. Overall, 304 patients and 61 HCWs were sampled. For each patient, the surface of the infected surgical site was cleaned with normal saline, before swabbing took place. Before the sample was collected, the area was rinsed with sterile normal saline, then, a sterile cotton swab was gently rolled over the surface of the wound. The swab with pus was kept in a sterile test tube with a cap. HCWs participated in the study voluntarily, and nasal swab samples were collected from both of their anterior nares using sterile swabs. Each swab was rubbed against the anterior first 1 cm of the nasal vestibular wall. Metadata were collected at the time of the swab sampling through a standardized questionnaire regarding demographic variables, antibiotic use, and hospitalization in the last six months.

### 2.3. Bacterial Identification

The bacteriological analyses of samples were performed in a laboratory of bacteriology (Cotonou, Benin) using conventional methods, as described previously [24]. During transportation, the samples were stored in Amies transport medium at room temperature. The Patient wound swabs were inoculated on different types of agar media, including blood agar, Chapman agar, chocolate agar, and eosin methylene blue agar. They were then incubated at a temperature of 37 °C for 24 h. The plates were further re-incubated for up to 48 h in the case of no growth after 24 h before being reported as sterile. Further identification was performed regarding colony morphology, Gram staining, and biochemical testing for identification and speciation [25]. The nasal swab samples were plated onto Oxoid Brilliance MRSA medium (Oxoid, Ltd., Basingstoke, UK) aerobically at 37 °C and examined for growth after 24 and 48 h of incubation according to the manufacturer’s instructions. The growth of colonies showing blue coloration was indicative of MRSA. 

For both the patient and HCW samples, cefoxitin disc diffusion tests for predicting MRSA were performed, using 30 µg discs (Oxoid, UK) on Mueller-Hinton agar as described previously [26]. The colonies were stored at −80 °C in trypticase soy agar prior to molecular analysis. Furthermore, all identifications were confirmed in Belgium, using matrix-assisted laser desorption-ionization time-of-flight mass spectrometry (MALDI-TOF MS) according to the procedure described by Bizzini and Greub [27]. All the media were checked for their sterility. Strains of *Escherichia coli* ATCC (American Type Culture Collection, Manassas, VA, USA) 25922, and *S. aureus* ATCC 25923 were used as reference strains in the quality control of the antimicrobial susceptibility testing and other biochemical tests. 

### 2.4. Whole-Genome Sequencing

The DNA of the MRSA isolates was extracted following a protocol adapted from Unal and colleagues [28]. Single colonies were suspended in 50 µL of lysostaphin (0.1 mg/mL) (Sigma-Aldrich, Overijse, Belgium) and incubated at 37 °C for 10 minutes. Next, a mix of 45 µL of sterile water, 5 µL of proteinase K (2 mg/mL) (Sigma-Aldrich, Overijse, Belgium) and 150 µL of TRIS-HCL (0.1 M, pH 8) were added to the suspension, which was then incubated at 56 °C for 10 min, then at 95 °C for 5 min and centrifuged at 13,000 g for 5 minutes. The supernatants were recovered and stored at −20 °C until their further use. The DNA purity and concentration were assessed using the Nusanodrop 1000 (Isogen LifeScience, Utrecht, The Netherlands). The isolate sequencing libraries were created using Nextera XT DNA library preparation (Illumina, San Diego, CA, USA) according to the manufacturer’s instructions and were subsequently sequenced on a MiSeq instrument (Illumina, San Diego, CA, USA) using MiSeq V3 chemistry (Illumina) to produce 2 × 250 bp paired end reads. The sequencing protocol was repeated for some isolates because the generated data did not meet the minimum quality requirements, as indicated in Appendix A.

### 2.5. Quality Control and Preprocessing

The raw reads were trimmed using Trimmomatic 0.38 [29] with the following options: ‘LEADING’ set to 10, ‘TRAILING’ set to 10, ‘SLIDINGWINDOW’ set to ‘4:20’, ‘MINLEN’ set to 40, and ‘ILLUMINACLIP’ set to ‘NexteraPE-PE. fa: 2:30:10’. The processed reads were then de novo assembled using SPAdes 3.13.0 [30] with the ‘--careful’ option enabled and ‘--cov-cutoff’ set to 10. Contigs smaller than 1000 bp were removed using the seq function of Seqtk 1.3 (available at https://github.com/lh3/seqtk, accessed on 20 December 2022). The quality of the assemblies was evaluated using QUAST 4.4 [31], providing the filtered assemblies as input. The median depth was evaluated by mapping the processed reads against the filtered assembly using Bowtie2 2.4.1 [32] with the ‘--end-to-end’ and ‘--sensitive’ options enabled. Afterwards, the ‘depth’ function of SAMtools 1.9 [33] was utilized with the ‘-a’ option enabled to calculate the median depth. Kraken2 2.0.7 [34] was used to screen the datasets for contamination from other microbial species. The reads were classified using an in-house constructed database containing all the NCBI RefSeq ‘Complete genome’ entries (database accessed on the 11 February 2021) with the accession prefixes NC, NW, AC, NG, NT, NS, and NZ from the following taxonomic groups: archaea, bacteria, fungi, human, protozoa, and viruses. This database also contained a selection of birds, mammals, and arthropods reference genomes, which are listed in Appendix A. Datasets with >1% of reads assigned to a species outside of the *Staphylococcus* genus were considered as contaminated. Lastly, the datasets were screened for intra-species contamination using ConFindr 0.7.4 with the default settings [35]. Only datasets that were not contaminated (i.e., not flagged by Kraken2 or ConFindr), with a median depth over 25×, and a N50 over 10,000 bp were retained for the analysis.

### 2.6. Isolate Characterization

The online SCCmecFinder 1.2 tool [11] was used to check for the presence of the *mecA* gene and to determine the SCC*mec* types. Blastn 2.6.0 [36] was used with the default options to determine the *spa* types by aligning the assembled contigs against the *spa* repeat sequences collected from the Ridom *spa* server (available at https://spaserver.ridom.de/accessed on 20 December 2022). Sequence typing was performed as described previously [37] using the *S. aureus* MLST and core-genome MLST schemes retrieved from PubMLST.org (accessed 1 May 2022). The assembled contigs were screened for antimicrobial resistance (AMR) genes using the NCBI Antimicrobial Resistance Gene Finder (AMRFinder) tool 3.10.18 [38] with the ‘--coverage_min’ option set to 0.9, the ‘--organism’ option set to ‘*Staphylococcus aureus* and the database version ‘2021-12-21.1’. The presence of genes encoding virulence factors was evaluated using the BLASTN-based gene detection workflow described previously using the sequences from the Virulence Finder database (accessed 5 June 2022) [37,38,39]. Only hits with >90% target coverage and >90% sequence identity were retained. The integrity of the open-reading frames (ORFs) of the detected virulence genes was evaluated using GAMMA 1.4 [40] with the default settings. The sequences of these genes were extracted from the Virulence Finder database and harmonized by fixing the orientation (5′->3′), and removing all nucleotides upstream of the start codon and downstream of the stop codon prior to the analysis. As the AMR detection was based on translated nucleotide sequences, the completeness of these ORFs was not verified separately.

### 2.7. Phylogenomic Investigation

A core-genome MLST allele matrix was constructed by combining the typing outputs, only considering perfect hits (i.e., 100% covered and 100% nucleotide identity). Firstly, datasets with <90% of alleles identified were removed from the allele matrix. Afterwards, loci detected in <90% of datasets were removed. GrapeTree 2.2 [41] was then used to construct a minimum spanning tree from the filtered allele matrix with the method parameter set to ‘MSTreeV2’. Separate SNP-based phylogenies were constructed for all detected sequence types (STs) with multiple isolates from this study. SnapperDB [42] 1.0.6 and PHEnix 1-4 (available at https://github.com/phe-bioinformatics/PHEnix accessed on 20 December 2022) were used to call single nucleotide polymorphisms (SNPs) and calculate SNP addresses. The SNP address is a strain level seven-digit nomenclature based on the number of pairwise SNP differences. Each digit represents the cluster membership for the corresponding number of SNP differences, starting (right to left) with 0 (i.e., no SNP differences) to 5, 10, 25, 50, 100, and 250. Isolates sharing the same cluster digit differ by fewer than the corresponding number of SNPs from at least one other isolate in the corresponding cluster [42]. Reference genomes with matching STs were selected from the set of complete *S. aureus* genomes available in the NCBI assembly database. The sequences with accession numbers GCF_000013425.1, GCF_001444345.1, and GCF_008630695.1 were selected for ST8, ST121, and ST152, respectively. The SNPs located in regions marked as phages by the online version of PHASTER [43] were removed, then the VCF files were inserted into SnapperDB. The SNP matrices were extracted from SnapperDB using the ‘get_the_snps’ function. Maximum likelihood phylogenies were then constructed using MEGA 10.0.4 [44], which was also used to determine the most suitable nucleotide substitution model. The Kimura two-parameter model was selected for all three STs. The ‘Gaps/Missing data treatment’ option was set to ‘partial deletion’, the ‘Site coverage cutoff to 50%, the ‘Branch swap filter’ to ‘Very weak’, the ‘ML heuristic method’ to ‘SPR-3’, and the number of bootstrap replicates to 100. 

### 2.8. Ethics Approval and Consent to Participate

The study protocol was approved by the Ethics Committee of the Faculty of Health Sciences (FSS, Benin) under the reference number: 012-19/UAC/FSS/CER-SS. Written informed consent was obtained from both the patients and HCWs before their enrolment into the study. Confidentiality and personal privacy were respected at all stages of the study. All authors have read and agreed to the present version of manuscript.

## 3. Results

### 3.1. Sample Collection and Species Identification

An overview of the sample collection is provided in Figure 1. A total of 304 wound swabs were collected from patients with clinical signs of SSIs, from which 229 isolates were obtained. These comprised 49 *S. aureus* isolates, of which 65.3% (32/49) were MRSA. From HCWs, a total of 61 nasal swab samples were collected from which 15 (24.6%) MRSA isolates were collected. 

### 3.2. WGS Data Analysis

#### 3.2.1. Preprocessing and Data Quality Evaluation

In total, 33 high-quality WGS datasets were generated across six sequencing runs, corresponding to 27 MRSA isolates, as indicated in Appendix A. Due to the quality of the DNA extraction, which was not optimal due to limited resources, for 6 out of the 33 isolates, no high-quality WGS dataset could be generated, as indicated in Appendix A. When multiple datasets from one isolate passed all the QC checks, the least fragmented (i.e., based on having the highest N50 value) was retained for analysis. The raw and trimmed read counts for the 27 selected high-quality datasets are listed in Appendix A. The median number of read pairs before and after trimming was 457,715 and 423,759, respectively. An overview of the assembly statistics is provided in Appendix A, with a median N50 of 257,715 bp, and a median total assembly length of 2,850,650 bp. The sequencing depth ranged from 32× to 115×, with a median of 62×. 

#### 3.2.2. Isolate Typing

The STs, *spa* types, and SCC*mec* types for each isolate are shown in Appendix A. The 27 isolates were assigned to five STs. ST8 was the most prevalent (n = 15), and contained isolates collected from HCWs (n = 9), as well as patients (n = 6). ST121 and ST152 were both observed in five isolates, including one ST121 case in a HCW, and four others in patients. ST772 and ST789 were each only observed in a single isolate, collected from a patient and a HCW, respectively. Multiple *spa* types were observed among the isolates assigned to ST8 and ST152. For ST8, the isolates were assigned to t4176 (n = 13) and t121 (n = 2). For ST152, three different *spa* types were observed: t1096 (n = 2), t4690 (n = 2), and t355 (n = 1). The remaining *spa* types corresponded perfectly with STs: t657 (ST772), t091 (ST789), and t314 (ST121). The following SCC*mec* types were observed: IV (n = 13), IVa (n = 7), V (n = 2), Va (n = 2), Vc (n = 2), and XII (n = 1). For the thirteen isolates assigned to SCC*mec* type IV, the *ccrA2*, *dmecR1*, and *CCB2* genes were missing, and the prediction was based solely on the homology to the complete SCC*mec* cassette, which was covered for ~55%. For isolate 2656, no hits were found based on the homology of the whole cassette, and the prediction was based solely on the detected genes.

#### 3.2.3. AMR Prediction

An overview of the detected genes and mutations associated with AMR is provided in Table 1. A resistance to aminoglycosides was predicted for 16 isolates, due to the presence of the *aac*(6′)-Ie/*aph* (2″)-Ia (n = 13), *aph*(3′)-IIIa (n = 1), or both (n = 2) genes. A resistance to beta-lactam antibiotics was predicted for all isolates, due to the presence of the *mecA* gene. The other detected genes associated with a resistance to beta-lactam antibiotics were: *blaI* (n = 23), *blaR1* (n = 5), *blaZ* (n = 23), *mecI* (n = 1), and *mecR1* (n = 1). Except for isolate 1002 (ST789), all isolates were also predicted to be resistant to fosfomycin. All the ST8, ST121, and ST772 isolates carried the *fosB* gene. Additionally, point mutations associated with resistance to fosfomycin were found, including: *glpT* A100V (n = 10), *glpT* L27F (n = 5), *murA* E291D (n = 10), *murA G257D* (n = 2), and *murA* T396N (n = 10). A resistance to macrolides was predicted for four isolates which carried the *erm(C)* (n = 2), *mph(C)* (n = 2), or *msr(A)* (n = 2) genes. A resistance to quaternary ammonium was predicted for eight isolates (all ST8) that carried the *qacC* gene. Point mutations associated with resistance to quinolones, including *gyrA* S84L (n = 17), *parC* S80F (n = 1), and *parC* S80Y (n = 16), were detected in 17 isolates. Isolate 1401 was the sole isolate not carrying a functional *tet (38)* gene, as it contained a 1 bp deletion at position 215, resulting in an inactive gene product. Lastly, a resistance to trimethoprim was predicted for 24 isolates, due to the presence of the *dfrG* (n = 24) and *dfrS1* (n = 12) genes. 

#### 3.2.4. Virulence Gene Detection

The detected virulence genes are shown in Table 2. All isolates carried at least one virulence gene associated with the production of exo-enzymes, indicated in green in Table 2. Most isolates carried additional serine protease genes: *splA* (n = 16), *splB* (n = 18), and *splE* (n = 15). The five ST152 and the single ST772 isolate were the only isolates that, except for *aur*, lacked genes associated with exo-enzyme production. Except for isolate 2090 (ST8), all isolates also carried one or more of the following genes associated with host immunity (indicated in red in Table 2): arginine catabolic mobile element (ACME) (n = 2), *sak* (n = 25), and *scn* (n = 26). The third category, indicated in blue in Table 2, contained genes associated with the production of toxins. Four of the five ST152 isolates carried the *edinB* gene. All isolates carried the *hlgA* and *hlgB* genes, associated with the production of γ-hemolysin, and a subset of these isolates (n = 22) additionally carried the *hlgC* gene. The *lukF-PV* and *lukS-PV* genes, associated with the production of Panton–Valentine leukocidin, were detected in 13 isolates: the two ST8 isolates with *spa* type t121, the five ST121 isolates, the five ST152 isolates, and the one ST772 isolate. Fifteen isolates carried at least one staphylococcal enterotoxin gene (e.g., *sea*, *seb*…). Two of the detected staphylococcal enterotoxin genes carried mutations resulting in inactive gene products: the *seg* gene in isolate 5729 carried an SNP resulting in a premature stop-codon at codon 186 (of 259), and the *ser* gene in isolate 996 carried a single nucleotide deletion resulting in a frameshift at codon 60 (of 260). Lastly, the *tst* gene was observed in eight isolates, all assigned to ST8. We did not observe any association between the virulence gene profile and the origin of the isolates (e.g., collected from a patient or from a HCW).

#### 3.2.5. Phylogenomic Investigation

The core genome MLST allele matrix contained 1654 loci after filtering. The resulting minimum spanning tree is shown in Figure 2. The clades in the phylogeny correspond to the classic MLST classification. Within the same ST, isolates with different *spa* types were clearly separated. For ST152, the number of allelic differences between isolates with different *spa* types was in the 20–40 range, indicating that the core genomes of these isolates were relatively similar. Contrastingly, within ST8, the two isolates with *spa* type t121 differed by over 258 alleles from the other ST8 isolates. The clusters corresponding to ST8 and ST121 were the only ones that contained isolates obtained from HCWs as well as patients. The SNP-based phylogenies and corresponding SNP addresses for ST8, ST121, and ST152 are shown in Figure 3. For ST8, the SNP distances were relatively large. The two isolates with *spa* type t212 differed by over 250 SNPs from the thirteen t1476 isolates. A clade of four HCWs from different hospitals shared the same first three digits of the SNP address (i.e., the 25–50 SNP difference level), which was a subclade of the *spa* type t1476 clade, which shared the first two digits (i.e., the 50–100 SNP difference level). This larger clade contained isolates collected from nine HCWs and four patients, originating from the six hospitals that participated in this study. For ST121, relatively large distances existed between the isolates, as the SNP addresses only shared the first two digits (i.e., the 50–100 SNP difference level), comprising four patients and one HCW isolate from three hospitals. Despite the high genomic similarity (e.g., virulence genes, *spa* type), no signs of a very close phylogenetic relationship were observed between these isolates. For ST152, the topology of the SNP phylogeny corresponded to the *spa* type classification (i.e., isolates with the same *spa* type clustering together), comprising only patient samples from four hospitals. The two *spa* type t1096 isolates (905 and 907) were both collected from patients, and differed by only two SNPs, indicating that they are highly likely to be related infections, especially since they were collected in the same hospital, less than a year apart. The two *spa* type t4960 samples (isolates 2007 and 3247) were both collected from patients, and differed by 63 SNPs, coming from two different hospitals. Lastly, the single *spa* type t355 isolate shared only the first digit of the SNP address (i.e., the 250–100 SNP difference level).

## 4. Discussion

The characterization of circulating MRSA strains is essential to implementing efficient measures for their control and prevention. To the best of our knowledge, this is the first study to describe the genomic characterization of MRSA strains collected from hospitals in Benin. 

A high rate of MRSA in *S. aureus* collected from patients (32/49; 65.31%) and HCWs (15/61; 24.60%) was observed, much higher than the rate previously reported in hospitalized patients in Burkina Faso in 2016 (3.9%) [45]. This could indicate that MRSA has become much more widespread in Africa in recent years. These results are in accordance with studies conducted in Algeria, which showed a nearly tenfold rate of increase over 18 years, from less than 5% in 1996–1997 [46] to 50% in 2011 [47] and 62% in 2014 [48]. Because chromogenic media were used, the characterization of colonizing microorganisms was limited to MRSA, and, therefore, other species (such as methicillin-susceptible *S. aureus*) were potentially not identified. The isolates collected in this study were predicted to be resistant to multiple antibiotics. While the detection of genes and mutations associated with AMR does not necessarily correspond to phenotypic resistance, WGS has been shown to be an excellent predictor for AMR in *S. aureus* [49]. Most isolates were predicted to be resistant to oral antibiotics such as penicillin, tetracycline, fluoroquinolones, and cotrimoxazole (trimethoprim/sulfamethoxazole), all of which can be purchased without a prescription in Benin, potentially contributing to this high rate of resistance. Our findings are similar to those of a study conducted in Ghana in 2020, which reported similar rates of resistance to these antibiotics in a collection of 28 MRSA strains isolated from chronically infected wounds [50]. Many isolates in our study also carried the *dfrG* and *dfrS1* genes, associated with a resistance to trimethoprim (a component of cotrimoxazole), as observed elsewhere [51]. In Benin, cotrimoxazole is inexpensive, orally administered, and easily available. The high rate of predicted AMR to antibiotics commonly available in Benin highlights that awareness, education, and training to ensure responsible antibiotic use by patients, pharmacists, and physicians, are essential. 

One of the most important virulence factors of CA-MRSA is PVL, for which the encoding genes were frequently observed in our dataset. The high dissemination of PVL+ *S. aureus* in Sub-Saharan Africa has previously been linked to three major clones. The first was ST15, which was not observed in our dataset. The second and third, ST121 and ST152 [52,53], accounted for ten of the thirteen PVL+ strains in our dataset. The remaining three isolates carrying PVL genes were assigned to ST8 (n = 2) and ST772 (n = 1). Other studies, conducted in 2008 and 2012, have reported that ST152 was the most prevalent clone circulating in West Africa, accounting for 40–60% of infections, in both community and hospital settings [54,55]. Cases of this clone have also been reported in Europe [56,57], Turkey [58], and Haiti [59]. Another important virulence factor of *S. aureus* is the toxic shock syndrome toxin (TSST). The *tst* gene encoding TSST was detected in eight isolates in this study, all assigned to ST8, and collected from patients (n = 4) and HCWs (n = 4). As isolates co-harboring the *tst* gene and other toxin genes have been described as highly pathogenic [60], the spread of these pathogenic strains could pose a public health risk. However, as is the case for AMR, the presence of a virulence gene does not necessarily mean that the gene is expressed, and that the resulting toxin is created.

The phylogenomic investigation identified overall large distances between several clusters of potentially related isolates with highly similar core genomes. In some cases, we found a relatively high genomic similarity between isolates collected from HCWs and those from patients in hospitals, indicating the possibility of circulation between both groups. For example, the five isolates assigned to ST121 clustered together when collapsing branches at two cgMLST alleles. This potential cluster contained four isolates collected from patients and a single isolate collected from a HCW, originating from three different hospitals (Figure 3B). The SNP-based investigation showed that all samples differed between 50–100 SNPs, therefore finding no evidence of direct transmissions, but confirming that these isolates exhibit a high level of genomic relatedness. Similarly, the 13 isolates assigned to *spa*-type t1476 (ST8) exhibited similar core genomes, forming a single cluster when collapsing branches at five cgMLST alleles. The SNP analysis showed that the SNP distances varied from 25 to 100 SNPs, despite these 13 samples being collected from both patients and HCWs, from different hospitals, over a relatively long period. Lastly, within ST152, two strains from patients visiting the same hospital almost one year apart were identical except for two SNPs, indicating that these infections were very likely related and transmitted within the community. Although no direct signs of transmission between patients and HCWs were observed, potentially linked isolates may have been missed due to the relatively low number of isolates included in the WGS-based analysis (i.e., only a few isolates per hospital); meaning that the full diversity of strains circulating in the hospitals has not been captured, and the sampling strategy whereby patients and HCWs were sampled at different time points. Nevertheless, the large genomic similarity in terms of the core genome confirmed by the SNP-based analysis, and the high similarity in terms of AMR genes and virulence genes, between the strains collected from patients and HCWs in different hospitals (including samples taken in the same hospitals), suggest that strains colonizing HCWs may constitute a reservoir that can potentially result in the clinical infection of patients. 

The characterization of the colonizing isolates can offer valuable information regarding the risk of subsequent *S. aureus* infection, as nasal screening and decolonization in MRSA-positive patients have been found to substantially reduce the risk of SSIs [61]. As described elsewhere, the screening of HCWs is recommended when PVL-producing MRSA infections are observed in hospitals [62], when unusual epidemiological outbreak patterns are observed, or when a persistent MRSA carriage by HCWs is observed. Our study demonstrates that WGS constitutes an excellent method for the complete pathogen typing and characterization of MRSA, including the prediction of AMR, the detection of virulence genes, and determining the relatedness between strains. Considering the sub-optimal hand-hygiene compliance observed in hospitals in Benin [22], it is possible that HCWs contributed to the spread of MRSA in this context. While various studies have highlighted that screening and decolonization before certain types of surgery play crucial roles in reducing the rate of SSIs, proper hand hygiene remains a relatively easy preventive measure for reducing MRSA in LMICs. 

## 5. Conclusions

The severe complications that can result from MRSA SSIs pose a substantial burden on healthcare systems worldwide, especially in LMICs. Our study shows that MRSA strains carrying multiple AMR and virulence genes are circulating in hospitals in Benin, including genomically similar strains in both patients and HCWs, for which WGS and other surveillance methods can provide valuable information to help combat these infections and reduce the spread of AMR. 

## Figures and Tables

**Figure 1 microorganisms-11-01954-f001:**
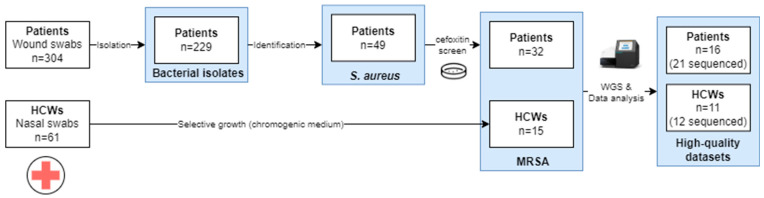
Overview of the sample collection approach for the samples to be used in the WGS. Abbreviations: healthcare worker, HCW. Only the high-quality datasets obtained after different runs are presented in the study.

**Figure 2 microorganisms-11-01954-f002:**
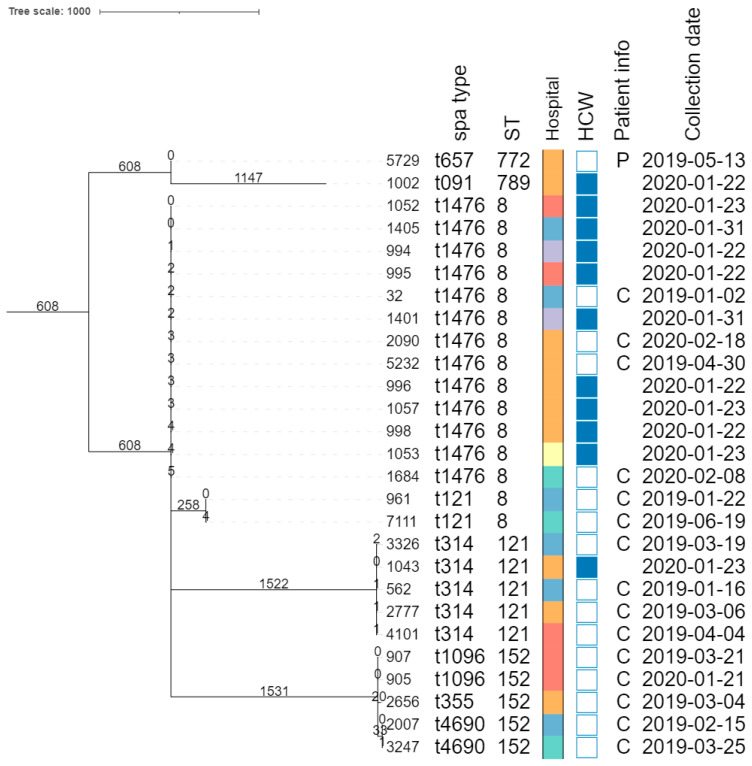
Core-genome MLST phylogeny. Branch lengths and the scale bar are expressed as the number of allele differences. The annotations are, from left to right: isolate name, *spa* type, sequence type (ST), hospital code (see legend at the bottom-left), patient vs HCW (a filled rectangle indicates that the corresponding isolate was collected from a HCW), patient information if available (P, peritonitis; C, cesarean), and collection date (yyyy-mm-dd). The six hospitals participating in the study were randomly assigned an identifier from A to F.

**Figure 3 microorganisms-11-01954-f003:**
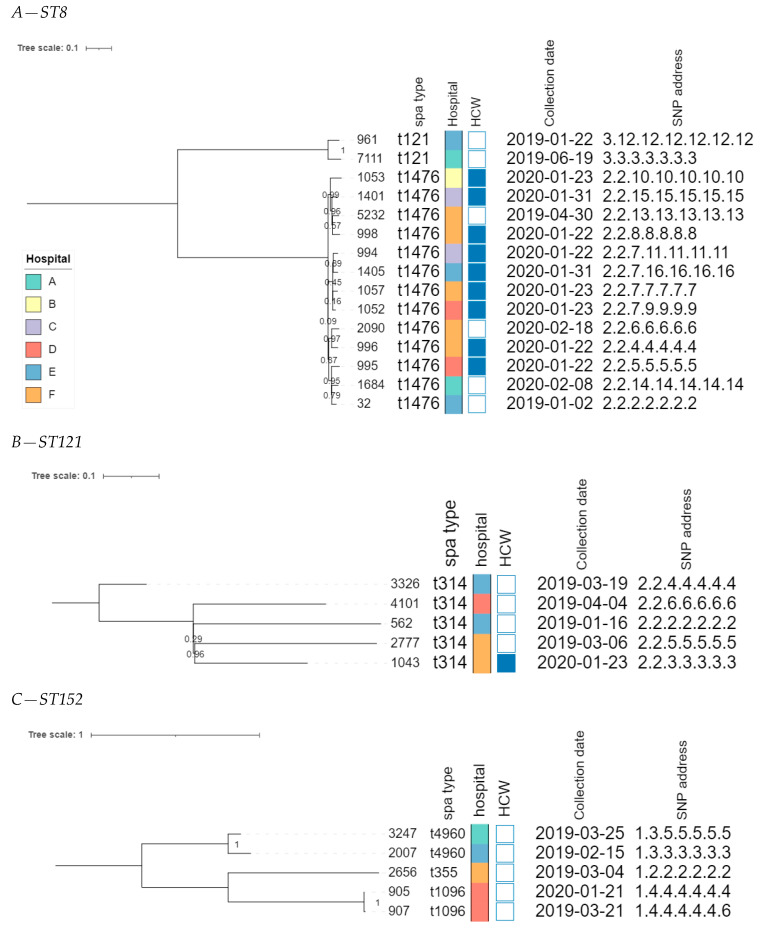
SNP-based phylogenies for ST8, ST121, and ST152. This figure shows the SNP-based phylogeny for ST8 (**A**), ST121 (**B**), and ST152 (**C**). Branch lengths and the scale bar are expressed as the average substitutions per site. The bootstrap values are shown next to the corresponding nodes. The annotations are, from left to right: isolate name, *spa* type, hospital, patient vs HCW (a filled rectangle indicates that the corresponding sample was collected from a HCW), collection date, and SNP address. The six hospitals participating in the study were randomly assigned an identifier from A to F, and are colored according to the legend depicted in panel A.

**Table 1 microorganisms-11-01954-t001:** The detected genes and mutations associated with AMR.

ST	*spa* Type	Isolate	AMGs		BLAs						FOF						MLs			QA	QNs			ST	TET		TMP	
			*aac(6′)-Ie aph(2″)-Ia*	*aph(3′)-IIIa*	*blaI*	*blaR1*	*blaZ*	*mecA*	*mecI*	*mecR1*	*fosB*	*glpT* A100V	*glpT* L27F	*murA* E291D	*murA* G257D	*murA* T396N	*erm(C)*	*mph(C)*	*msr(A)*	*qacC*	*gyrA* S84L	*parC* S80F	*parC* S80Y	*sat4*	*tet(38)*	*tet(K)*	*dfrG*	*dfrS1*
8	t121	7111	0	0	0	0	0	1	0	0	1	0	0	0	1	0	0	0	0	0	1	0	1	0	1	0	0	0
8	t121	961	0	1	1	1	1	1	0	0	1	0	0	0	1	0	0	1	1	0	1	0	1	1	1	0	0	0
8	t1476	1684	1	0	1	0	1	1	0	0	1	0	0	0	0	0	0	0	0	1	1	0	1	0	1	0	1	1
8	t1476	2090	1	0	1	0	1	1	0	0	1	0	0	0	0	0	1	0	0	1	1	0	1	0	1	1	1	1
8	t1476	32	0	0	1	0	1	1	0	0	1	0	0	0	0	0	0	0	0	1	1	0	1	0	1	1	1	1
8	t1476	5232	1	0	1	0	1	1	0	0	1	0	0	0	0	0	0	0	0	0	1	0	1	0	1	1	1	1
8	t1476	998	1	0	1	0	1	1	0	0	1	0	0	0	0	0	0	0	0	0	1	0	1	0	1	1	1	1
8	t1476	1052	1	0	1	0	1	1	0	0	1	0	0	0	0	0	0	0	0	1	1	0	1	0	1	0	1	1
8	t1476	1053	1	0	1	0	1	1	0	0	1	0	0	0	0	0	0	0	0	1	1	0	1	0	1	0	1	1
8	t1476	1057	1	0	1	0	1	1	0	0	1	0	0	0	0	0	0	0	0	0	1	0	1	0	1	1	1	1
8	t1476	1401	1	0	1	0	1	1	0	0	1	0	0	0	0	0	1	0	0	0	1	0	1	0	0	1	1	1
8	t1476	1405	1	0	1	0	1	1	0	0	1	0	0	0	0	0	0	0	0	1	1	0	1	0	1	0	1	1
8	t1476	994	1	0	1	0	1	1	0	0	1	0	0	0	0	0	0	0	0	0	1	0	1	0	1	1	1	0
8	t1476	995	1	0	1	0	1	1	0	0	1	0	0	0	0	0	0	0	0	1	1	0	1	0	1	1	1	1
8	t1476	996	1	0	1	0	1	1	0	0	1	0	0	0	0	0	0	0	0	1	1	0	1	0	1	0	1	1
121	t314	3326	0	0	1	0	1	1	0	0	1	1	1	1	0	1	0	0	0	0	0	0	0	0	1	0	1	0
121	t314	2777	0	0	1	0	1	1	0	0	1	1	1	1	0	1	0	0	0	0	0	0	0	0	1	1	1	0
121	t314	4101	0	0	1	0	1	1	0	0	1	1	1	1	0	1	0	0	0	0	0	0	0	0	1	0	1	0
121	t314	562	0	0	1	0	1	1	0	0	1	1	1	1	0	1	0	0	0	0	0	0	0	0	1	1	1	0
121	t314	1043	0	0	1	0	1	1	0	0	1	1	1	1	0	1	0	0	0	0	0	0	0	0	1	0	1	0
152	t1096	905	0	0	1	1	1	1	0	0	0	1	0	1	0	1	0	0	0	0	0	0	0	0	1	1	1	0
152	t1096	907	0	0	1	1	1	1	0	0	0	1	0	1	0	1	0	0	0	0	0	0	0	0	1	1	1	0
152	t355	2656	1	0	0	0	0	1	1	1	0	1	0	1	0	1	0	0	0	0	0	0	0	0	1	1	0	0
152	t4690	2007	0	0	0	0	0	1	0	0	0	1	0	1	0	1	0	0	0	0	0	0	0	0	1	1	1	0
152	t4690	3247	0	0	0	0	0	1	0	0	0	1	0	1	0	1	0	0	0	0	0	0	0	0	1	1	1	0
772	t657	5729	1	1	1	1	1	1	0	0	1	0	0	0	0	0	0	1	1	0	1	0	1	0	1	0	1	0
789	t091	1002	1	1	1	1	1	1	0	0	0	0	0	0	0	0	0	0	0	0	1	1	0	1	1	1	1	0

Overview of the detected genes and mutations associated with AMR. The first, second, and third columns list the STs, *spa* types, and isolate names, respectively. The subsequent columns indicate the presence of the corresponding gene or mutation(s). ‘1’ indicates that the feature was detected, and ‘0’ indicates that it was absent. Features associated with AMR were grouped based on the classes defined in the AMRFinder database, as indicated by the top row. Abbreviations: sequence type, ST; aminoglycosides, AMGs; ß-lactamases, BLAs; bleomycin, BLE; fosfomycin, FOF; macrolides, MLs; quaternary ammonium, QA; quinolones, QNs; streptothricin, ST; tetracycline, TET; trimethoprim, TMP.

**Table 2 microorganisms-11-01954-t002:** Detected virulence genes.

ST	*spa* Type	Isolate	Exo-Enzymes				Host Immunity			Toxins																							
			*aur*	*splA*	*splB*	*splE*	*ACME*	*sak*	*scn*	*edinB*	*hlgA*	*hlgB*	*hlgC*	*lukD*	*lukE*	*lukF-PV*	*lukS-PV*	*sea*	*seb*	*sec*	*seg*	*sei*	*sej*	*sek*	*sel*	*sem*	*sen*	*seo*	*sep*	*seq*	*ser*	*seu*	*tst*
8	t121	7111	1	1	1	1	1	1	1	0	1	1	1	1	1	1	1	0	0	0	0	0	0	1	0	0	0	0	0	1	0	0	0
8	t121	961	1	1	1	1	1	1	1	0	1	1	1	1	1	1	1	0	0	0	0	0	0	0	0	0	0	0	0	0	0	0	0
8	t1476	1684	1	1	0	1	0	1	1	0	1	1	1	1	1	0	0	0	0	0	0	0	1	0	0	0	0	0	0	0	1	0	1
8	t1476	2090	1	1	1	1	0	0	0	0	1	1	1	1	1	0	0	0	0	0	0	0	0	0	0	0	0	0	0	0	0	0	1
8	t1476	32	1	1	0	0	0	1	1	0	1	1	1	1	1	0	0	0	0	0	0	0	1	0	0	0	0	0	0	0	1	0	1
8	t1476	5232	1	1	1	1	0	1	1	0	1	1	1	1	1	0	0	0	0	0	0	0	0	0	0	0	0	0	0	0	0	0	1
8	t1476	998	1	1	1	1	0	1	1	0	1	1	1	1	1	0	0	0	0	0	0	0	0	0	0	0	0	0	0	0	0	0	1
8	t1476	1052	1	1	1	1	0	1	1	0	1	1	1	1	1	0	0	0	0	0	0	0	1	0	0	0	0	0	0	0	1	0	0
8	t1476	1053	1	1	1	1	0	1	1	0	1	1	1	1	1	0	0	0	0	0	0	0	1	0	0	0	0	0	0	0	1	0	1
8	t1476	1057	1	1	1	1	0	1	1	0	1	1	1	1	1	0	0	0	0	0	0	0	1	0	0	0	0	0	0	0	1	0	0
8	t1476	1401	1	1	1	1	0	1	1	0	1	1	1	1	1	0	0	0	0	0	0	0	1	0	0	0	0	0	0	0	1	0	1
8	t1476	1405	1	1	1	1	0	1	1	0	1	1	1	1	1	0	0	0	0	0	0	0	0	0	0	0	0	0	0	0	0	0	0
8	t1476	994	1	1	1	1	0	1	1	0	1	1	1	1	1	0	0	0	0	0	0	0	1	0	0	0	0	0	0	0	1	0	0
8	t1476	995	1	1	0	1	0	1	1	0	1	1	1	1	1	0	0	0	0	0	0	0	0	0	0	0	0	0	0	0	0	0	1
8	t1476	996	1	1	1	1	0	1	1	0	1	1	1	1	1	0	0	0	0	0	0	0	1	0	0	0	0	0	0	0	0 ^(2)^	0	0
121	t314	3326	1	0	1	0	0	1	1	0	1	1	1	1	1	1	1	0	1	0	1	1	0	0	0	1	0	1	0	0	0	1	0
121	t314	2777	1	0	1	0	0	1	1	0	1	1	1	1	1	1	1	0	1	0	1	1	0	0	0	1	1	1	0	0	0	1	0
121	t314	4101	1	0	1	0	0	1	1	0	1	1	1	1	1	1	1	0	1	0	1	1	0	0	0	1	1	1	0	0	0	1	0
121	t314	562	1	0	1	0	0	1	1	0	1	1	1	1	1	1	1	0	1	0	1	1	0	0	0	1	1	1	0	0	0	1	0
121	t314	1043	1	0	1	0	0	1	1	0	1	1	1	1	1	1	1	0	1	0	1	1	0	0	0	1	1	1	0	0	0	1	0
152	t1096	905	1	0	0	0	0	1	1	1	1	1	0	0	0	1	1	0	0	0	0	0	0	0	0	0	0	0	0	0	0	0	0
152	t1096	907	1	0	0	0	0	1	1	1	1	1	0	0	0	1	1	0	0	0	0	0	0	0	0	0	0	0	0	0	0	0	0
152	t355	2656	1	0	0	0	0	1	1	0	1	1	0	0	0	1	1	0	0	0	0	0	0	0	0	0	0	0	0	0	0	0	0
152	t4690	2007	1	0	0	0	0	1	1	1	1	1	0	0	0	1	1	0	0	0	0	0	0	0	0	0	0	0	0	0	0	0	0
152	t4690	3247	1	0	0	0	0	1	1	1	1	1	0	0	0	1	1	0	0	0	0	0	0	0	0	0	0	0	0	0	0	0	0
772	t657	5729	1	0	0	0	0	0	1	0	1	1	1	0	0	1	1	1	0	1	0 ^(1)^	1	0	0	1	1	1	1	1	0	0	1	0
789	t091	1002	1	1	1	1	0	1	1	0	1	1	1	1	1	0	0	0	0	0	0	0	0	0	0	0	0	0	0	0	0	0	0

Overview of the detected virulence genes. The first, second, and third columns list the STs, *spa* types, and isolate names, respectively. The subsequent columns indicate the presence of the corresponding gene. A ‘1’ indicates that the gene was detected, and ‘0’ indicates that the gene was absent. Colors correspond with the three categories of virulence gene, as defined in the Virulence Finder [38] database, shown in the top row of the table. Abbreviations: sequence type, ST. Notes: (1) the *seg* gene in isolate 5729 contained a mutation at position 557, resulting in a premature stop-codon (codon 186 of 259), and was, therefore, considered absent; (2) the *ser* gene in isolate 996 contained a single nucleotide deletion at position 124, resulting in a frameshift (codon 60 of 260) and was, therefore, also considered absent.

## Data Availability

The datasets supporting the conclusions of this study have been deposited in the NCBI SRA, under the accession number PRJNA936823. Individual accession numbers are provided in Appendix A. Data access for reviewers is provided through the following link: https://dataview.ncbi.nlm.nih.gov/object/PRJNA936823?reviewer=sf152q21290p7jkk213b8p9rq0, accessed on 20 December 2022.

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
