# Peer review of "Whole-Genome Sequencing-Based Screening of MRSA in Patients and Healthcare Workers in Public Hospitals in Benin"

_microorganisms, 2023, doi:10.3390/microorganisms11081954_

Round 1

Reviewer 1 Report

The authors present their findings on whole genome analysis of MRSA isolates from healthcare workers and patients across six different hospitals. The work was conducted adequately and the findings are sound but I have the following concerns or items that need to be addressed:

1. The abstract needs to be rewritten as it is misleading and confusing. It gives the reader the impression that this study analysed 365 MRSA isolates (306 + 61), when in fact there were only 27. Furthermore, the abstract claims that all isolates carried the mecA gene, when the data in the paper only indicates this for the 27 isolates that underwent whole genome sequencing. Furthermore the percentages supplied in the abstract are also confusing as these are in reference to the 27 isolates and not 365 as alluded to earlier.

2. Line 75 contains a short sentence that appears to be unfinished.

3. The method for whole genome sequencing is rather crude. DNA should be cleaned via either ethanol precipitation or column purification prior to library preparation to ensure high quality output. 

4. The paper is a little ambiguous regarding the quality of the data sets. While in some places it refers to the data as "high-quality" in other parts of the paper it refers to using the "least fragmented" and Tables S2 and S3 refer to the data as "OK". 

5. 61 samples were collected from healthcare workers but it isn't clear if S. aureus was isolated from all of these. The only data provided is that there were 15 MRSA. What about the other 46 samples? Was methicillin susceptible S. aureus isolated from the other 46? 

6. The overall datasets are quite small but the authors have adequately addressed this limitation in the discussion.

Author Response

Dear Reviewer, Please find in attachement the responses to all comments. 

Reviewer 2 Report

The main issue of the study is to observe whether whole genome sequencing of S. aureus obtained from infected surgical sites and from nasal swabs of hospital staff have similar characteristics. This finding may suggest pathways for nosocomial infections.

Healthcare-associated infection (HAI) is a topic of global concern. Whole genome analysis has become an important tool in molecular epidemiology, investigating at the same time resistance genes, biofilm-forming genes, virulence genes, as well as plasmids and mutations. Thus, combining this tool with such an impactful theme makes the study relevant.

This study comparatively reveals the routes of dissemination of the infection not only in the country, but even in the continent. Despite the obvious conclusion that hand hygiene care is essential to considerably reduce nosocomial infection, characteristics of the bacteria (Sts, PVL...) circulating in that niche were discussed.  

To explore broader databases, our group   submits assembled sequences to the CARD database directly through their website. Our samples are classified by SCCmec and protein A (spA) typing, using staphtopia and spatyper, respectively. It is just an alternative, since the methodology chosen in the case of this study is efficient.

On the other hand, we were interested to know that KRAKEN was used to investigate pollution or contamination. We consider evaluating this option to improve our work.

I consider the study to be of great value. Whole-genome sequencing is a valuable tool; however, it is not simple to perform and even less to analyze. The costs are high, and we are still learning how to make the most of the information that is available. The data were treated with great care, and the phylogeny was used very correctly. I congratulate the group and the editors.

My only suggestion is that you review the numbers provided in the abstract.

I did not identify relevant errors in the use of the English language

Author Response

Dear reviewer 2, Please see in attachement the responses to your comment.

regards 
